# Two's company, three species is a crowd? A webcam-based study of the behavioural effects of mixed-species groupings in the wild and in the zoo

Claire Gauquelin Des Pallieres[1], Paul E. Rose [1,2]*

1 Centre for Research in Animal Behaviour, Psychology, University of Exeter, Exeter, Devon, United Kingdom, 2 WWT, Slimbridge Wetland Centre, Slimbridge, Gloucestershire, United Kingdom

* p.rose@exeter.ac.uk

## Abstract

Mixed species exhibits in zoos are used to create larger, more stimulating environments to support naturalistic interactions between species. In the wild, mixed species groups are observed as having lower rates of vigilance, presumably due to reduced predation risk through 'detection' and 'dilution' effects. This effect appears to be highly variable depending on factors such as food availability or degree of threat. This study aimed to collect data on mixed-species associations and consequent vigilance rates in the wild, collecting equivalent data from a large mixed-species zoo enclosure to compare the findings between free-ranging and captive populations. The study additionally investigated whether large mixed-species enclosures support natural associations and behaviours, by comparing the behaviour of captive animals with wild counterparts. The study used livestream video feeds from 10 national parks in South Africa and Kenya to observe free-ranging species, and a camera at the San Diego Zoo Safari Park's mixed species African exhibit. Scan and continuous sampling protocols were used simultaneously to record behavioural states as well as the rate of scanning (vigilance) events. GLMMs were run to test whether vigilance of a focal species varied according to the number of animals present, the density of animals in the group, and the diversity of species. In the wild, vigilance decreased with increasing number of animals in the surroundings but in captivity the group size had no impact. The results suggest that in the wild, these species benefit from increased perceived safety in larger groups, regardless of the species making up that group. No effect was noted in the zoo because of a reduced need for animals to show heightened vigilance to the same degree as in the wild. Similarities were observed in associations between species/mixed species group compositions, and in behaviour budgets. These findings provide a preliminary evaluation of how the impact of mixed species groupings may translate from the wild to the zoo, based on the associations and behaviour across a variety of African ungulates.

**Data Availability Statement:** All relevant data are within the paper and its Supporting Information files.

**Funding:** The author(s) received no specific funding for this work.

**Competing interests:** The authors have declared that no competing interests exist.

## 1. Introduction

Captive environments frequently provide insufficient space and behavioural stimulation for many animals [1,2]. These restrictions can lead to behavioural issues ranging from apathy, which manifests as low levels of activity and lowered positive behavioural diversity [3], to abnormal repetitive behaviours [4,5] that are often linked to impoverished welfare states [6]. Increasingly, naturalistic and more stimulating environments are required to keep improving standards of welfare in the zoo [7], and to conserve adaptive, species-specific behaviours in zoo populations which are then more suitable for (any potential) reintroductions into the wild [8]. Mixed-species enclosures allow space for the creation of larger and more socially enriched environments [9], significantly more akin to natural habitats. However, mixed-species enclosures also pose risks, such as aggression between species [10] and a potential for chronic stress to occur [11]. Nonetheless, these risks can often be avoided by housing biologically relevant species mixes in combination with careful management of each species housed [9]. The most successful species combinations tend to be naturally occurring combinations [12]. Therefore, it is important to develop a strong understanding of species associations in the wild and how these compare to those seen within zoo enclosures. This way mixed-species enclosure development can be guided by these natural associations and aim to benefit the welfare of all species involved. It is also important to compare the behavioural impact of species associations across both wild and captive environments and evaluate the extent to which a mixed species group supports natural associations and behaviours. Specifically, we must understand what animals associate in the wild and why this may be, and how these associations might change and influence behaviour under different conditions (e.g. resource access or degree of perceived threat) [13].

Mixed-species groups are observed across a huge range of species, including ungulates [14], primates [15], fish [16], and birds [17]. Such groupings provide multiple adaptive benefits, such as efficient foraging and predator defence [18]. Further benefits occur to animals via associated behavioural changes, such as through reduced rates of vigilance, due to increased perceived safety from predation, which allows animals to spend more time on important behaviours such as feeding [19] and drinking [20]. Evidence suggests that mixed-species groups can similarly provide significantly positive behavioural and welfare influences in captivity [12]. For example, research on mixed-species groups in captivity found that various primate species, such as capuchin (*Cebus apella*) and squirrel monkeys (*Saimiri sciureus*) showed increased activity levels and behavioural diversity, as well as reduced conspecific aggression [21]. Little research has investigated the influence of mixed-species groupings on the behaviour of non-primate species in the zoo even though this needs consideration, as positive outcomes and negatives outcomes of different species mixes are noted [9,13]. Additionally, few studies have investigated the influence of mixed bird and mammal groupings on behaviour patterns and degree of vigilance, even though these types of animals can be housed together in zoo enclosures [22]. It is also important to consider how other social and environmental factors, such as habitat type, sex, and presence of predators [23] impact on the composition and formation of mixed species groups in the wild, and therefore how any impacts of species mixing may translate to a zoo environment and to the behaviours displayed by an enclosure's occupants.

This study aimed to collect data on natural species associations in the wild, and the behavioural changes associated with different species assortment. Additionally, this study collected equivalent data from a zoo enclosure, to compare these findings between zoo and wild. The study focussed on a wide range of East and South African species of birds and herbivorous mammals. The study aimed to answer four main questions, comparing these outcomes

between captive and wild conspecifics: (1) What animals associate with each other in the wild and in the zoo? (2) How does the presence of other species influence rates of vigilance? (3) What social factors influence the likelihood of forming a mixed species group and associated vigilance rates? (4) How does habitat type influence the formation of mixed species groups and associated vigilance rates?

The study used live-streamed cameras to observe wild animals found at waterholes across South Africa and Kenya, as well as a camera focussed on captive animals at a San Diego Zoo Safari Park mixed species exhibit. The study used the proximity of animals to each other as an indicator of association, labelling animals within six body lengths of each other as being within a mixed species group [22,24]. This was particularly important for the environments in this study (i.e., zoo enclosures and waterholes) because animals experienced a certain degree of forced proximity due to shared resource use at waterholes and being in the same enclosure in the zoo, and thus animals in the same area (i.e., visible in the camera in the vicinity) were not necessarily choosing to associate [24,25]. Six body lengths were deemed appropriate based on previous studies [22,24] and our own pilot observations because whilst animals are using the same resource in both environments, there was enough space for individuals to maintain significant distance from one another. These data showed a visible contrast between species that stayed within six body lengths of each other and species that were simply present in the same area (e.g., in view of the waterhole) but did not come close to each other (such as elephants and impala). Together, these data would enable understanding of how associations between species and associated vigilance rates vary between wild and zoo environments and additionally allow a comparison of the behaviour of animals housed in a large mixed species enclosure with their wild counterparts.

It was hypothesised that:

i. Vigilance rates would be lower when animals were in mixed-species groups, due to increased perceived safety from predators through a dilution effect [26] or from the effect of "many eyes" hypothesis [27].

ii. Vigilance was expected to decrease when group size increased [28] and when groups contained mainly juvenile animals due to lack of experience of predation risk [29].

iii. Animals would engage in fewer intraspecific aggressive behaviours when they were in mixed-species groups [21] as an increased choice of associates would be available.

iv. Animals in captivity would have lower vigilance rates overall compared to wild counterparts due to reduced predation risk [30].

v. The impacts of mixed species grouping on behaviour would be consistent between the wild and the zoo.

## 2. Methods

### 2.1. Sample population

Data were collected using live-streamed webcams located at national parks throughout South Africa and Kenya, and at the San Diego Zoo Safari Park. These cameras were obtained from three websites: www.explore.org, www.africam.com and www.sdzsafaripark.org (see *Appendix A* in S1 File for further information on each camera's location and individual website link). Eight of the wild camera locations were in the north-east of South Africa, located at six different game reserves and national parks: Balule National Park, Pilanesberg National Park, Tembe Elephant Park, Madikwe National Park, Olifants West Game Reserve, and the Sabi Sands

Game Reserve. The other three wild camera locations were based in Kenya's Laikipia County. The final camera was based at the "African Plains" exhibit at the San Diego Zoo Safari Park in California, USA. The outdoor exhibit at the safari park is roughly 12-hectares in area, containing nine species of ungulate: Cape buffalo (*Syncerus caffer*), southern white rhinoceros (*Ceratotherium simum*), Masai giraffe (*Giraffa camelopardalis tippelskirchi*), fringe-eared oryx (*Oryx beisa callotis*), defassa waterbuck (*Kobus ellipsiprymnus*), Nile lechwe (*Kobus leche*), impala (*Aepyceros melampus*), blue wildebeest (*Connochaetes taurinus*), Grant's gazelle (*Nanger granti*), and Thompson's gazelle (*Eudorcas thomsonii*); and several bird species. The enclosure consists of various habitat spaces including areas of vegetative cover, a waterhole feature, and areas of open grassland. The camera is based at the "Kijami Overlook", mid-way along the length of the enclosure, and can pan sideways and zoom, allowing visibility across a large proportion of the exhibit [31].

Throughout the study, data were collected on a total of 43 species in 388 observations across both wild and captive environments. This included 18 species of ungulate and elephant, two species of primates, seven species of bird, one species of cat, and one species of crocodile (Table 1). For wild data collection, all mixed species presence was recorded (i.e. observations of predatory species in proximity with prey species) even if such occurrences would not be replicated in the zoo. For final analyses, only zoo-relevant mixed species occurrences (i.e. those that would be feasible in captivity) were included. Using the zoo webcam, we successfully collected data on eight captive species: giraffe, fringe-eared oryx, Nile lechwe, defassa waterbuck, white rhinoceros, impala, Cape buffalo, and blue wildebeest. Of these eight captive species, six were also observed in the wild (giraffe, defassa waterbuck, white rhinoceros, impala, Cape buffalo, and blue wildebeest).

This study received ethical approval from the CLES Psychology Ethics Committee on 17th March 2021 (eCLESPsy002406). As a courtesy, the San Diego Zoo was contacted by email on 19th January 2021 in advance of the study to inform them of the planned project and idea for use of the African Plains webcam for data collection. All data collection for the study made use of publicly available live-streamed webcams freely available on the San Diego Zoo Safari Park, africam and explore.org websites, and we did not violate any of the terms of use for these websites.

## 2.2. Procedure

Data collection took place four days a week between 2nd June and 11th August 2021, during the daylight hours for each camera location (between 07:30 and 18:30 local time). Focal samples of ten minutes were carried out, using both all-occurrence and instantaneous sampling methods [32]. Cameras were checked on an hourly basis, and the first species sighted in the centre of the screen was chosen as the focal species. Observations were ended if the focal species went out of sight, and only samples of five minutes or longer were used in analyses.

**2.2.1. Interspecific associations.**   We recorded the presence of all other species within the same 'group' as the focal species. A group was defined as all animals within six body lengths of each other [33,34]. As a large variety of species were observed, the body length of a mid-sized antelope, in this case the nyala at 1.35 to 1.95 metres, was used for all focal species. These six body lengths were estimated visually as a circumference around the focal animal by comparing to the length of the focal species present on-screen. If any individual of a species was within 6 body lengths of the nearest focal individual, the group was labelled as a mixed species group. Fig 1 illustrates how a group was defined.

**2.2.2. Vigilance levels.**   We used two different methods to record vigilance levels both as a proportion of time and as a frequency. The number of vigilance bouts performed by the focal species was recorded using all-occurrence sampling [32] throughout the 10-minute period.

**Table 1. The species that were observed during the study are shown below, along with the number of focal observations for that species.**

| Class | Family | Species | No. focal obs. |
|---|---|---|---|
| Mammalia | Bovidae | **Impala (*Aepyceros melampus*)** ** | 57 |
| | | **Nyala (*Tragelaphus angasii*)** | 32 |
| | | **Greater kudu (*Tragelaphus strepsiceros*)** | 9 |
| | | Sitatunga (*Tragelaphus spekii*) | 1 |
| | | **Cape bushbuck (*Tragelaphus sylvaticus*)** | 12 |
| | | **Defassa waterbuck (*Kobus ellipsiprymnus*)** ** | 29 |
| | | **Nile lechwe (*Kobus leche*)** * | 13 |
| | | **Blue wildebeest (*Connochaetes taurinus*)** ** | 23 |
| | | Hartebeest (*Alcelaphus buselaphus*) | 1 |
| | | **Fringe-eared oryx (*Oryx beisa callotis*)** * | 10 |
| | | **Cape buffalo (*Syncerus caffer*)** ** | 9 |
| | | Common eland (*Taurotragus oryx*) | 1 |
| | | Grey rhebok (*Pelea capreolus*) | 1 |
| | | Kirk's dik-dik (*Madoqua kirkii*) | 1 |
| | Cercopithecidae | Vervet monkey (*Chlorocebus pygerythrus*) | 6 |
| | | Chacma baboon (*Papio ursinus*) | 5 |
| | Elephantidae | **African elephant (*Loxodonta africana*)** | 15 |
| | Equidae | **Plains zebra (*Equus quagga*)** | 22 |
| | Felidae | Lion (*Panthera leo*) | 2 |
| | Giraffidae | **Giraffe (*Giraffa camelopardalis*)** ** | 37 |
| | Rhinocerotidae | **White rhinoceros (*Ceratotherium simum*)** ** | 24 |
| | Suidae | **Warthog (*Phacochoerus africanus*)** | 16 |
| Aves | Anatidae | **Egyptian goose (*Alopochen aegyptiaca*)** | 20 |
| | | White-faced whistling duck (*Dendrocygna viduata*) | 2 |
| | | Spur-winged goose (*Plectropterus gambensis*) | 1 |
| | Ardeidae | Grey heron (*Ardea cinerea*) | 2 |
| | Ciconiidae | **Woolly-necked stork (*Ciconia episcopus*)** | 10 |
| | Scopidae | Hamerkop (*Scopus umbretta*) | 2 |
| | Struthionidae | Common ostrich (*Struthio camelus*) | 3 |

Those species coloured in grey were not included in the final analyses due to a low number of observations. * refers to a species observed in the zoo and ** refers to a species observed both in the zoo and in the wild.

This number was later divided by the length of the recording, and the average number of focal species present, to obtain a frequency of vigilance bouts. The behavioural states–i.e. long duration behaviours [32]–of the focal species were instantaneously sampled every minute using an ethogram (Table 1) developed from various sources [35–38]. The number of individuals performing each behaviour type was recorded at each minute interval. This was then divided by the average number of individuals present and the recording length to calculate the proportion of time an individual spent in a vigilant state. To simplify the final analyses, state behaviours were reduced to nine key behaviours and a category for 'other' (see 'category for analysis' in Table 2).

**2.2.3. Social factors.** Information about the social environment was noted before the start of each observation. The presence of juveniles was noted when they were clearly identifiable by their smaller size. The sex of individual animals was also noted for species where sexual dimorphism was identifiable on camera (e.g., for species of Bovidae such as for impala where males have horns whereas females do not). This resulted in categorisation of all-male, all-female, or

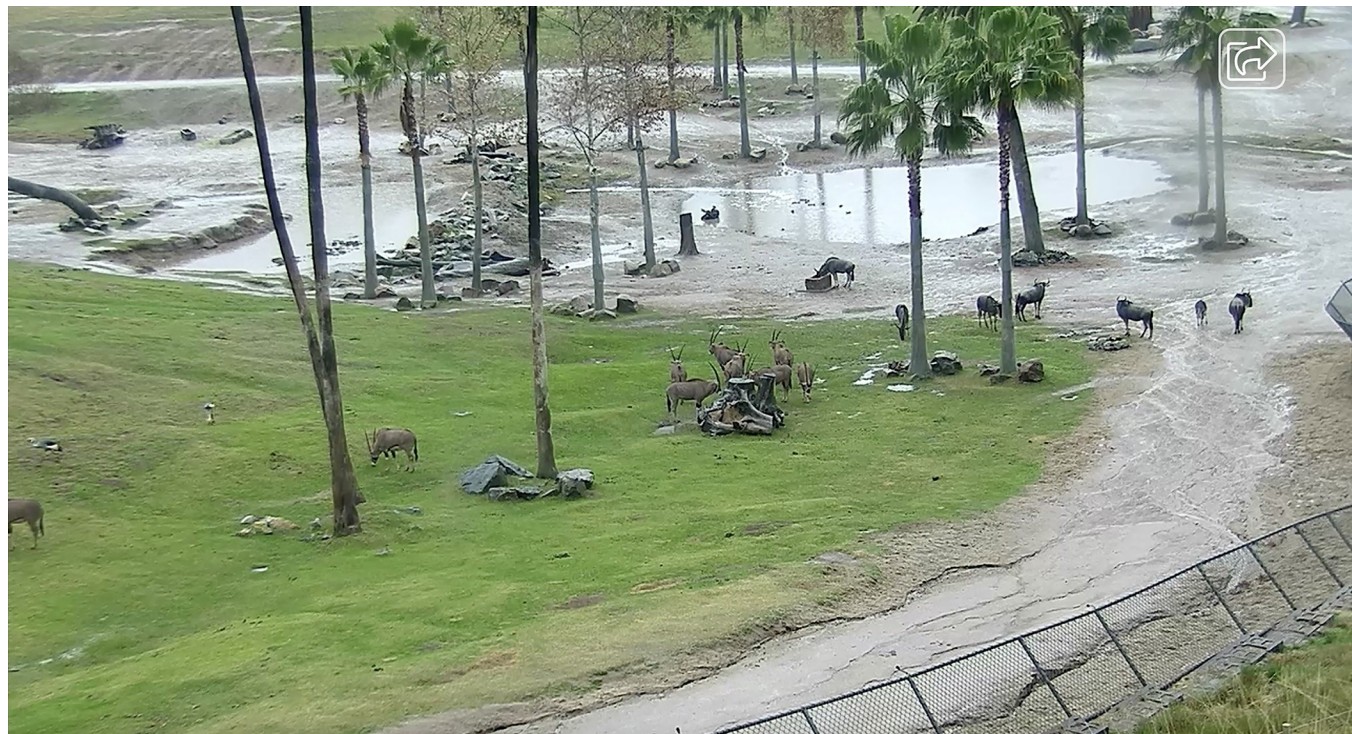

**Fig 1. An example of associations between oryx and wildebeest from the San Diego Zoo Safari Park's "African Plains" exhibit.** Screenshot of webcam footage.

mixed-sex groups. Overall, 150 observations in the study included data on the sex composition of the group.

Species diversity was determined as the number of different species present in the environment, along with the total number of individual animals present in the same environment. The species density was determined from the number of individuals within the same group as the focal individual. Additionally, we noted the body size of the focal species, categorised as small <100kg, medium 150 and 900kg, large at >1000kg, as body size influences predation risk [39] and vigilance [40]. Species were also categorised into feeding type (carnivorous mammals, browsing mammals, grazing mammals, intermediate feeding mammals, herbivorous birds, and carnivorous birds). Browsers, grazers and intermediate feeders for ungulates were based on Hofmann and Stewart [41]. A disturbance was recorded as present if a predator became visible on camera during an observation, or if there was human disturbance such as construction or cars driving through the camera view.

**2.2.4. Habitat type.**   The camera location and habitat type were recorded before the observation began. The surrounding habitat was labelled according to whether animals were at a waterhole (within the open area immediately surrounding a waterhole), on open grassland (using short grass plains away from a waterhole), or within under vegetative cover (surrounded by vegetation such as shrubs, or under cover from trees). This distinction was made for both wild and captive environments.

## 2.3. Statistical analysis

Data were analysed using Generalized Linear Mixed Models (GLMMs) in R v.4.1.1 [42] and RStudio v.1.1.463 [43]. We first ran a Chi-squared test of independence to investigate whether

**Table 2. Ethogram displaying how behaviours were defined, coded, and grouped for analyses.**

| Behavioural Type | Category for Analysis | Behaviour | Definition | Code |
|---|---|---|---|---|
| Event | Vigilance | Vigilance | Animal interrupts its current behaviour to raise its head above its shoulders, and scan or fixate environment. | n/a |
| | Affiliative Interaction | Interacting (affiliative) | Animals are making gentle body contact, moving in a coordinated matter, grooming each other, or playful. | Aff |
| | Aggressive Interaction | Interacting (aggressive) | Animals are shoving, chasing, or fighting each other. Animals may charge at, bite, or shove one another in an aggressive or violent manner. | Agg |
| State | Vigilance | Vigilant | Animal has its head upright above its shoulders, its ears perked, and its eyes fixated on the horizon, not doing anything else such as chewing. | Vig |
| | Feeding | Grazing | Animals have their heads lowered to the ground and are consuming shrubs or grass from the floor. Grazing in shallow water is included. | Gra |
| | | Browsing | Animals are consuming from a tree or tall brush such that their head and neck are upright and extended forwards. | Bro |
| | | Feeding | Animals are consuming from buckets or troughs on the ground filled with food prepared by keepers. | Fee |
| | | Chewing | Animals take a break during feeding to raise their head and chew food. | Che |
| | Drinking | Drinking | Animal is standing by a water source and have their head and neck lowered, consuming water through their mouth or trunk. | Dri |
| | Rumination | Rumination | Animal is chewing regurgitated food | Rum |
| | Resting | Standing still | Animal is standing on all legs however its head and neck are at level with their shoulders or lower, clearly not actively focussed on their environment. | Sta |
| | | Lying vigilant | Animal has its front body rested on the floor, but with its head upright and eyes focused on the horizon. | Lyi |
| | | Resting | Animal has its front body and head rested on the floor. | Res |
| | | Sleeping | Animal has its eyes closed for more than 5 seconds whilst either standing or lying down. | Sle |
| | Movement | Walking | Animals are slowly moving around on all legs in a relaxed gait. | Wal |
| | | Running | Animal is rapidly placing legs one in front of the other to carry itself forward rapidly. | Run |
| | | Wading in water | Animal is within a waterhole or river. It may be still or walking. This includes a bird or duck which is swimming. | Wad |
| | Self-Maintenance | Self-maintenance | This definition includes any behaviour where animals are directing behaviour at their own bodies. Animals may be grooming or preening themselves using their mouths or beaks, or scratching/cleaning their bodies by rubbing against their foot, a tree, or rubbing themselves on the ground, or otherwise washing themselves in water. | SM |
| | Other | Other | Any behaviour that does not fit with the above definitions (e.g., aggressive interactions that occurred between the focal species and another species). | Oth |

the number of times animals were observed in a mixed species group varied between the zoo and the wild. We used location type (zoo or wild) and species grouping (single or mixed) as binary categorical variables. We ran the same testing for four commonly observed species (both free living and under captive conditions) with social structures that could be markedly different between the zoo and the wild–giraffe (multilevel, matrilineal societies [44,45]), Cape buffalo (fission-fusion system very large mean herd size [46,47]), defassa waterbuck (male territory defence, female home range ownership [48] and blue wildebeest (million strong herd size, migratory populations [49])–to identify any potential impacts of zoo management on available social choices and therefore herd composition. We then ran a final chi-squared test of independence to assess whether the number of times animals were observed in a mixed species group varied between species. We then created a table to display the species associations observed throughout the study (Table 4).

To analyse the impact of mixed species grouping on vigilance, we first ran two Wilcoxon signed-rank tests to look at the overall difference in vigilance rates between single and mixed

**Table 4. Number of associations between two species, i.e., the number of times these two species were observed within the same 'group' (within 6 body lengths of each other) during one of their focal observations.**

| Species 1 | Species 2 | Frequency of Association |
|---|---|---|
| **Impala** | **Nyala** | **16** |
| | **Plains zebra** | **12** |
| | **Defassa waterbuck** ** | **7** |
| | **Blue wildebeest** | **6** |
| | **Egyptian goose** | **3** |
| | Chacma baboon | 3 |
| | **White rhinoceros** * | **3** |
| | **Giraffe** | **3** |
| | **Nile lechwe** * | **2** |
| | Ostrich | 2 |
| | Kirk's dik-dik | 2 |
| | Vervet monkey | 2 |
| | **African elephant** | **2** |
| **Zebra** | **Impala** | **12** |
| | **Blue wildebeest** | **10** |
| | **Warthog** | **5** |
| | **Egyptian goose** | **3** |
| | Vervet monkey | 2 |
| | Ostrich | 2 |
| **White rhinoceros** | **Nile lechwe** * | **13** |
| | **Cape buffalo** * | **5** |
| | **Defassa waterbuck** * | **5** |
| | **Fringe-eared oryx** * | **5** |
| | **Blue wildebeest** * | **3** |
| | **Impala** * | **3** |
| | **Giraffe** * | **2** |
| **Blue wildebeest** | **Zebra** | **10** |
| | **Impala** | **6** |
| | **White rhinoceros** * | **3** |
| | **Egyptian goose** | **3** |
| **Egyptian goose** | Red-knobbed coot | 3 |
| | **Plains zebra** | **3** |
| | **Impala** | **3** |
| | **Blue wildebeest** | **3** |
| | **Blacksmith plover** | **3** |
| | **Nyala** | **2** |
| **Nyala** | **Impala** | **16** |
| | **Defassa waterbuck** | **10** |
| | **African elephant** | **8** |
| | **Egyptian goose** | **2** |
| | **Woolly-necked stork** | **2** |
| **Nile Lechwe** | **White rhinoceros** * | **13** |
| | **Defassa waterbuck** * | **3** |
| | **Cape buffalo** * | **2** |
| | **Impala** * | **2** |

(*Continued*)

**Table 4.** (Continued)

| Species 1 | Species 2 | Frequency of Association |
|---|---|---|
| **Defassa waterbuck** | **Nyala** | **10** |
| | **Impala ** ** | **7** |
| | **White rhinoceros *** | **5** |
| | **Nile lechwe *** | **3** |
| **Giraffe** | **Fringe-eared oryx *** | **10** |
| | **Impala** | **3** |
| | **White rhinoceros *** | **2** |
| **Elephant** | **Nyala** | **8** |
| | **Woolly-necked stork** | **5** |
| | **Impala** | **2** |
| **Oryx** | **Giraffe *** | **10** |
| | **White rhinoceros *** | **5** |
| **Woolly-necked stork** | **African elephant** | **5** |
| | **Nyala** | **2** |
| **Cape buffalo** | **White rhinoceros *** | **5** |
| | **Nile lechwe *** | **2** |
| **Warthog** | **Plains zebra** | **5** |
| **Greater Kudu** | **Impala** | **2** |
| Hamerkop | Hippopotamus | 2 |
| White-faced whistling duck | Grey heron | 2 |

Associations not emboldened are of species not included in the final analyses. The * highlights species observed in the zoo and **highlights species observed both in the zoo and in the wild.

species groups within both zoo and wild environments separately. We then fitted four GLMMs with negative binomial error distribution and log link function [50]. The first GLMMs were run to investigate the frequency of vigilance events, and the further two were run to investigate the proportion of time spent vigilant. The seven fixed predictors originally included in the models were body size, feeding type, habitat type, time of day, disturbance, species diversity, species density, and total animals present. A backwards selection procedure was then run, using likelihood-ratio tests (LRT) to remove non-significant predictors until only significant ones were left for the final model. Then Akaike's Criterion (AIC) was used to compare models with and without each additional predictor. Six further Wilcoxon signed-rank tests were additionally run to compare the proportion of time that animals spent performing different key behaviours between wild and zoo environments, such as feeding, ruminating, and resting.

In all four negative binomial GLMMs mentioned above, we included the observation length and the average number of animals present in the observation, as offset terms to control for variation in observation effort [50]. We used the package *"car"* with function "vif" [51] to determine the variance inflation factor (VIF) as a test for the assumption of collinearity. There was one case of collinearity between feeding type and body size in the analysis of vigilance rate in the wild (VIF = 3.379), therefore the predictor of feeding type was removed. The rest of the analyses revealed no problems of collinearity (maximum calculated VIF of 1.781).

To analyse how social factors influenced the likelihood of forming a mixed species group we ran a GLMM with a logit link function and binomial error distribution [50]. This test only used wild data because this test required different groups of the same species. Two fixed

predictors were included in the original model (presence of juveniles and sex composition of group). A backwards selection procedure was then run, using LRT tests to remove non-significant predictors until only significant ones were left in the final model.

To investigate if habitat type influenced the likelihood of forming mixed species groups in the wild and in the zoo, we ran two binary logistic regressions. We used habitat type as a categorical predictor and mixed species grouping as a binary outcome variable. An LRT test was used to determine whether habitat type significantly increased the likelihood of forming a mixed species group.

Lastly, to investigate whether intraspecific aggression was less frequent in a mixed species group, we ran a binary logistic regression. For this model we used species grouping (single or mixed) as a binary predictor variable and the presence or absence of intraspecific aggression as a binary outcome variable. An LRT test was used to determine whether species grouping significantly predicted the likelihood of aggression occurring.

## 3. Results

### 3.1. Interspecific associations

Overall, the animals in this study spent 49.6% of their time in a mixed-species group in the wild, and 61.4% of their time in a mixed-species group in the zoo. This difference was not significantly different ($\chi^2$ = 2.88, df = 1, p = 0.090). For species that were considered individually, there was also no significant difference in the amount of time each species spent in a single-species or mixed-species group in the wild when compared to in the zoo (Table 3). Time spent in a mixed-species group overall varied significantly between species, from 16.4% for bushbuck to 84.6% for Nile lechwe, ($\chi^2$ = 33.13, df = 15, p = 0.005). See *Appendix B* in S1 File for output for each species. The mean species diversity observed in the wild was 2.45 species (SD = ± 0.0987), and in the zoo this was 2.65 (SD = ± 0.116 SD). The mean species density in the wild was 11.63 (SD = ± 0.714) and in the zoo this was 11.81 (SD = ± 1.063). Some species were found to have particularly high rates of association (Table 4), for example nyala and impala.

### 3.2. Vigilance

The rate of vigilance was significantly higher in the wild (0.77 scans per minute) compared to the zoo (0.25 scans per minute), (W(326) = 15464, p < 0.001). In the wild, there was no significant difference in vigilance rates between single (0.82) and mixed (0.72) species groups (W (237) = 7492, p = 0.372). The same was true for the zoo, with a mean vigilance of 0.27 in a single species group and 0.24 in a mixed species group, (W(87) = 964, p = 0.877).

The final model for the rate of vigilance in wild species included seven fixed predictors (species diversity, total animals present, time of day, disturbance, habitat type, and body size, N = 237). This was significantly better than the null model ($\chi^2$ = 67.613, df = 11, p < 0.001, $r^2$ = 30%). Vigilance rate was significantly influenced by species diversity ($\chi^2$ = 4.960, p = 0.030), habitat type ($\chi^2$ = 7.132, p = 0.028), time of day ($\chi^2$ = 5.828, p = 0.212), total animals present ($\chi^2$ = 38.271, p < 0.001), and body size ($\chi^2$ = 38.808, p < 0.001). The presence of disturbance

**Table 3. Time that each species spent in a mixed-species group in the wild compared to the zoo, assessed using Chi-squared tests of independence.**

| Species | $\chi^2$ | df | P value |
|---|---|---|---|
| Giraffe | 0.482 | 1 | 0.488 |
| Waterbuck | 0.392 | 1 | 0.531 |
| Wildebeest | 0 | 1 | 1.00 |
| Buffalo | 8.914e-33 | 1 | 1.00 |

($\chi^2$ = 3.275, p = 0.070; Table 5) did not have a significant impact on vigilance rates. Fig 2 illustrates the impact of species diversity on vigilance rates.

The final model for the rate of vigilance in zoo species included four fixed predictors (disturbance, species density, habitat type, and body size, N = 89). This was significantly better than the null model ($\chi^2$ = 31.583, df = 6, p < 0.001, $r^2$ = 20.0%). Vigilance rate was significantly influenced by body size ($\chi^2$ = 10.233, p = 0.006), disturbance ($\chi^2$ = 5.670, p = 0.017), but not species density ($\chi^2$ = 2.563, p = 0.109) or habitat type ($\chi^2$ = 2.695, p = 0.096; Table 6).

The final model for the proportion of time spent vigilant in the wild included four fixed predictors (disturbance, species diversity, species density, and body size, N = 237). This was significantly better than the null model ($\chi^2$ = 100.560, df = 5, p < 0.001, $r^2$ = 12.3%). The amount of time animals spent vigilant was significantly influenced by species density ($\chi^2$ = 4.679, p = 0.031), species diversity ($\chi^2$ = 5.368, p = 0.021), body size ($\chi^2$ = 7.310, p = 0.026), and disturbance ($\chi^2$ = 8.361, p = 0.004; Table 7).

The final model for the proportion of time spent vigilant in the zoo included two fixed predictors (body size and feeding type, N = 89). This was significantly better than the null model ($\chi^2$ = 37.959, df = 4, p < 0.001, $r^2$ = 19.8%). The amount of time animals spent vigilant was significantly influenced by feeding type ($\chi^2$ = 14.85, p = 0.001) but not body size ($\chi^2$ = 5.752, p = 0.056; Table 8).

Fig 3 illustrates the vigilance rate of individual species when in a single-species group compared to when other species were present with which they were frequently observed. In addition, Fig 4 illustrates how the sex composition of a group influenced vigilance rates in single and mixed species groups.

## 3.3. Other behaviours

There was no significant difference in the amount of time spent feeding in the wild (37.0%) compared to the zoo (33.1%), ($W(327)$ = 10583, p = 0.962). Animals spent more time drinking

**Table 5. Coefficients and p-values for the negative binomial GLMM investigating the influence of habitat type, feeding ecology, body size, time of day, total animals present, and species diversity on rates of vigilance in the wild.**

| Fixed effect | Estimate | Standard Error | P value |
|---|---|---|---|
| **Intercept** (no disturbance, waterhole, 07:00–09:00, small, grazer) | 0.104 | 0.249 | 0.676 |
| **Disturbance** | | | |
| Present | 0.332 | 0.215 | 0.124 |
| **Habitat** | | | |
| Open grassland | -0.395 | 0.170 | 0.020* |
| Vegetative cover | -0.302 | 0.160 | 0.059 |
| **Time of observation** | | | |
| 09:00–11:00 | 0.215 | 0.244 | 0.379 |
| 11:00–13:00 | 0.158 | 0.248 | 0.523 |
| 13:00–15:00 | -0.209 | 0.266 | 0.431 |
| 15:00–17:00 | 0.020 | 0.288 | 0.946 |
| **Body size** | | | |
| Medium | -0.282 | 0.140 | 0.044* |
| Large | -1.276 | 0.190 | <0.001* |
| **Population characteristics** | | | |
| Total animals present | -0.040 | 0.006 | <0.001* |
| Species diversity | 0.108 | 0.049 | 0.026* |

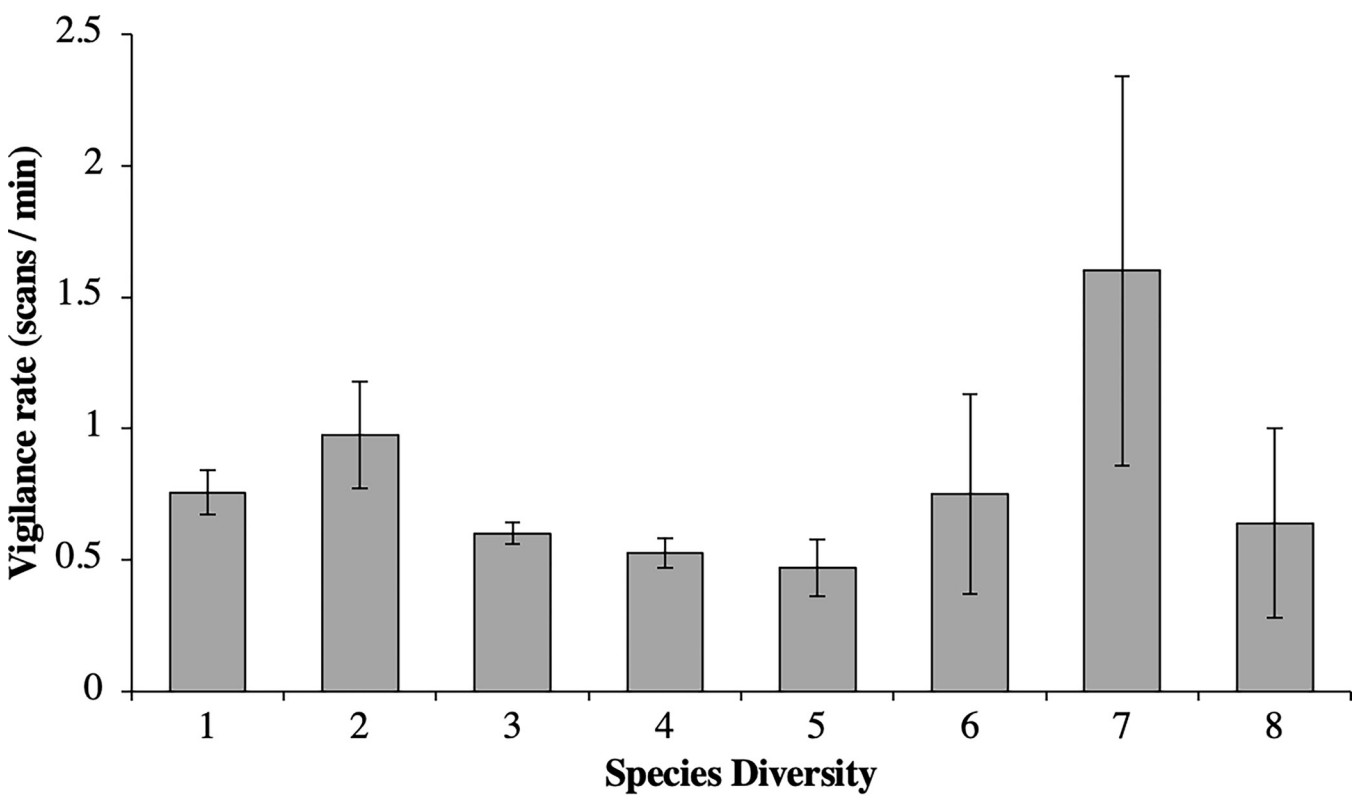

**Fig 2. Mean rate of vigilance (number of scans per minute) across different species diversities (number of different species present in the environment).**

in the wild (11.1%) than in the zoo (1.4%), (W(327) = 13760, p < 0.001). Less time was spent resting in the wild (5.4%) compared to the zoo (28%), (W(327) = 6619.5, p < 0.001), and more time locomoting in the wild (25.6%) than the zoo (8.5%), (W(327) = 12854, p = 0.002). More time was spent on self-maintenance in the wild (5.3%) than in the zoo (1.1%), (W(327) = 12004, p = 0.021). There was no difference in the proportion of time spent vigilant in the wild (19.9%) compared to the zoo (19.0%), (W(327) = 10750, p = 0.788). More time was spent affili-ating in the zoo (2.7%) than in the wild (1.4%), (W(327) = 8559, p < 0.001) but there was no

**Table 6. Coefficients and p-values for the negative binomial GLMM investigating the influence of habitat type, feeding ecology, body size, time of day, total animals present, and species diversity on rates of vigilance in the zoo.**

| Fixed effect | Estimate | Standard Error | P value |
|---|---|---|---|
| **Intercept** (no disturbance, waterhole, small) | 0.130 | 0.556 | 0.815 |
| **Disturbance** | | | |
| Present | 0.977 | 0.457 | 0.032* |
| **Habitat** | | | |
| Open grassland | -0.707 | 0.436 | 0.105 |
| Vegetative cover | -0.137 | 0.537 | 0.798 |
| **Body Size** | | | |
| Medium | -0.789 | 0.333 | 0.018* |
| Large | -1.106 | 0.351 | 0.002* |
| **Population characteristics** | | | |
| Species density | -0.023 | 0.012 | 0.066 |

**Table 7. Coefficients and P values for the negative binomial GLMM investigating the influence of habitat type, feeding ecology, and body size on the amount of time spent vigilant in the wild.**

| Fixed effect | Estimate | Standard Error | P value |
|---|---|---|---|
| **Intercept** (no disturbance, waterhole, small) | 0.130 | 0.556 | 0.815 |
| **Disturbance** | | | |
| Present | 0.689 | 0.154 | <0.001* |
| **Population characteristics** | | | |
| Species diversity | 0.118 | 0.053 | 0.025* |
| Species density | -0.014 | 0.007 | 0.049* |
| **Body Size** | | | |
| Medium | 0.453 | 0.172 | 0.008* |
| Large | 0.272 | 0.226 | 0.230 |

difference in aggressive interactions with 0.4% of time engaging in aggression in the wild and 0.5% in the zoo (W(237) = 10040, p = 0.057). Finally, there was no difference in rumination behaviour between wild (4.3%) and zoo (7.1%), (W(223) = 4574.5, p = 0.051).

### 3.4. Social factors

The factorial logistic regression to investigate how social factors influenced the formation of mixed species groups in the wild found no significant effects of the presence of juveniles or the sex of the focal species. Animals were not more or less likely to form mixed-species groups when juveniles were present (56%) or absent (53.2%) ($\chi^2$ = 0.020, p = 0.885), nor when the sex of the focal group was all-female (54.8%), all-male (50%), or mixed (55.8%), ($\chi^2$ = 0.010, p = 0.995; Fig 5; Table 9).

### 3.5. Habitat type

In the wild, animals were observed 154 times at a waterhole, 38 times on open grassland, and 45 times within vegetative cover. In the zoo, animals were observed six times at a waterhole, 73 times on open grassland, and 11 times under vegetative cover (Fig 6). The first factorial logistic regression for habitat type found this to have a significant effect on the formation of mixed species groups in the wild ($\chi^2$ = 7.748, p = 0.021; Fig 6; Table 10). The second regression found no significant effect of habitat type on the formation of mixed species groups in the zoo ($\chi^2$ = 2.000, p = 0.368; Fig 6).

**Table 8. Coefficients and P values for the negative binomial GLMM investigating the influence of habitat type, feeding ecology, and body size on the amount of time spent vigilant in the zoo.**

| Fixed effect | Estimate | Standard Error | P value |
|---|---|---|---|
| **Intercept** (grazer, small) | -2.211 | 0.328 | <0.001* |
| **Feeding ecology** | | | |
| Browser | 1.060 | 0.356 | 0.003* |
| Intermediate | 1.501 | 0.410 | 0.029* |
| **Body size** | | | |
| Medium | 0.628 | 0.384 | 0.102 |
| Large | -0.061 | 0.410 | 0.881 |

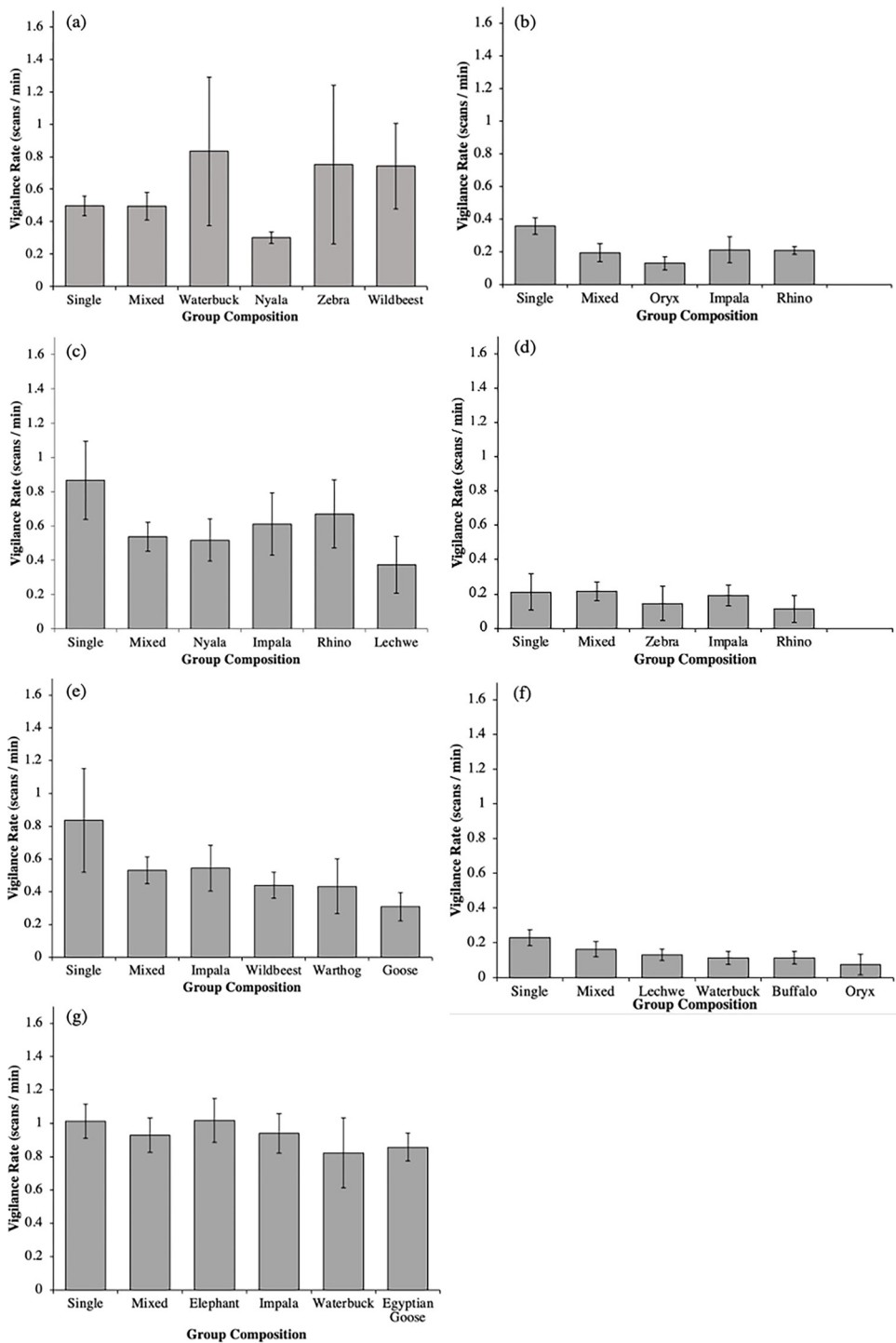

**Fig 3.** Rate of vigilance of (a) Impala (b) Giraffe (c) Defassa waterbuck (d) Wildebeest (e) Zebra (f) White rhinoceros and (g) Nyala, when in single and mixed species groups, or in the presence of another particular species.

### 3.6. Intraspecific aggression

A binary logistic regression found no significant difference in the rate of aggression between mixed and single species groups ($\chi^2$ = 0.835, p = 0.361).

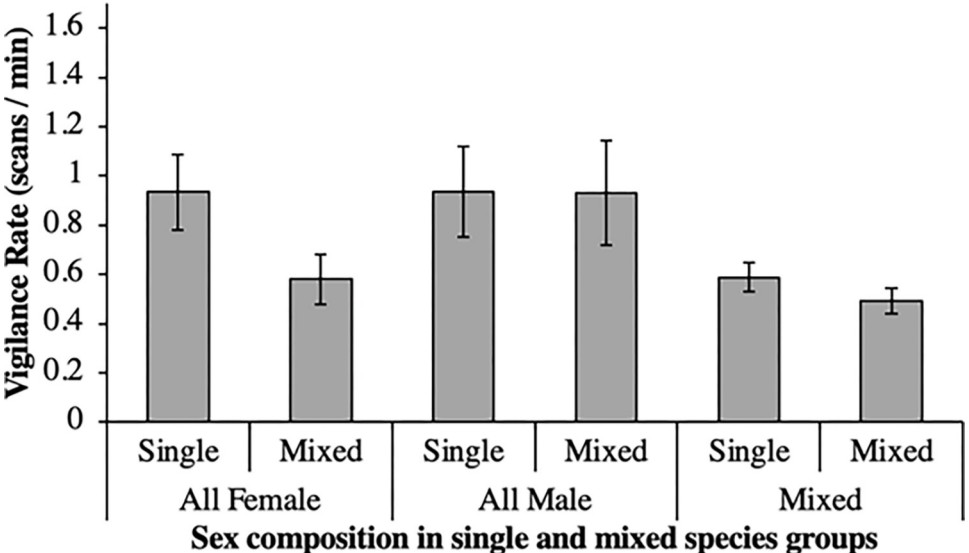

**Fig 4. The rate of vigilance when animal groups of different sex compositions were observed across single and mixed species groups.**

## 4. Discussion

This research has identified significant similarities in the association patterns and group compositions of zoo-housed individuals and their wild counterparts, supporting the suggestion that mixed-species enclosures provide more naturalistic and socially enriched environments. Captive individuals and their wild counterparts additionally demonstrated relatively similar activity patterns, despite some differences in the amount of time spent drinking and locomoting or resting, suggesting that this mixed species enclosure promoted species-typical activity patterns. The composition of animals within the focal group and in the surrounding

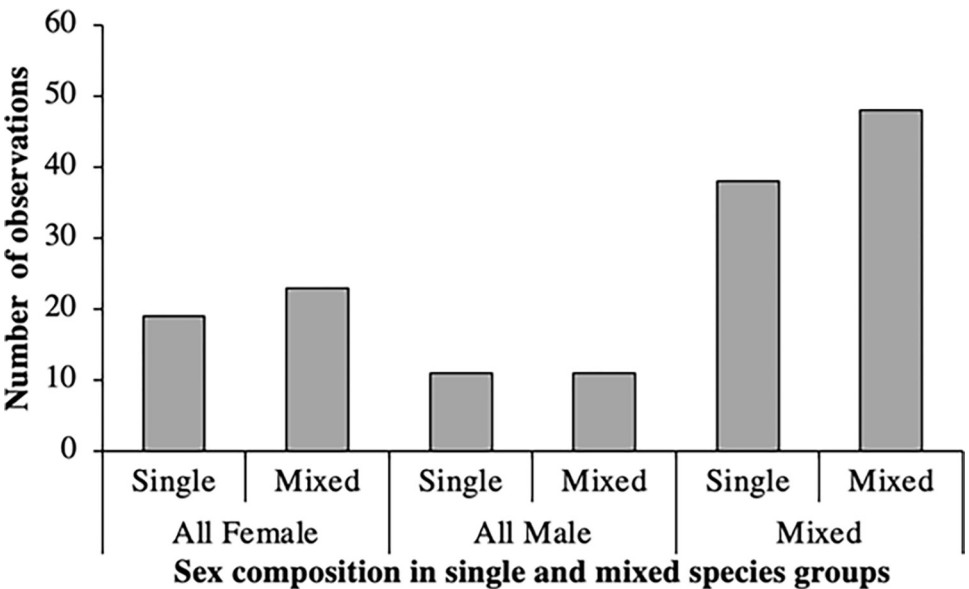

**Fig 5. Number of times animal groups of different sex compositions were observed in single and mixed species groups.**

**Table 9. Coefficients and P values for the final analysis on social factors influencing the formation of mixed-species groups (N = 126).**

| Fixed Effect | Estimate | Standard Error | P value |
|---|---|---|---|
| **Intercept** (no juveniles, all female group, focal herd size: 1) | -0.031 | 0.367 | 0.934 |
| **Presence of Juveniles** | | | |
| Juveniles present | -0.058 | -0.404 | 0.885 |
| **Sex Composition** | | | |
| All male | 0.031 | 0.527 | 0.957 |
| Mixed sex | -0.023 | 0.420 | 0.957 |

environment significantly influenced the vigilance of species in the wild, but there was no such effect in the zoo. There initially appear to be similarities between zoo and wild environments in how habitat type impacts the formation of mixed species groups and associated vigilance rates, although more data are required for this particularly from the zoo. The sex of the species and the presence of juveniles had no impact on the formation of mixed-species groups and associated vigilance rates.

## 4.1. Interspecific associations

This study observed high rates of association between species, with animals spending approximately half (52.5%) of their time in a mixed species group both in the wild and in the zoo. In

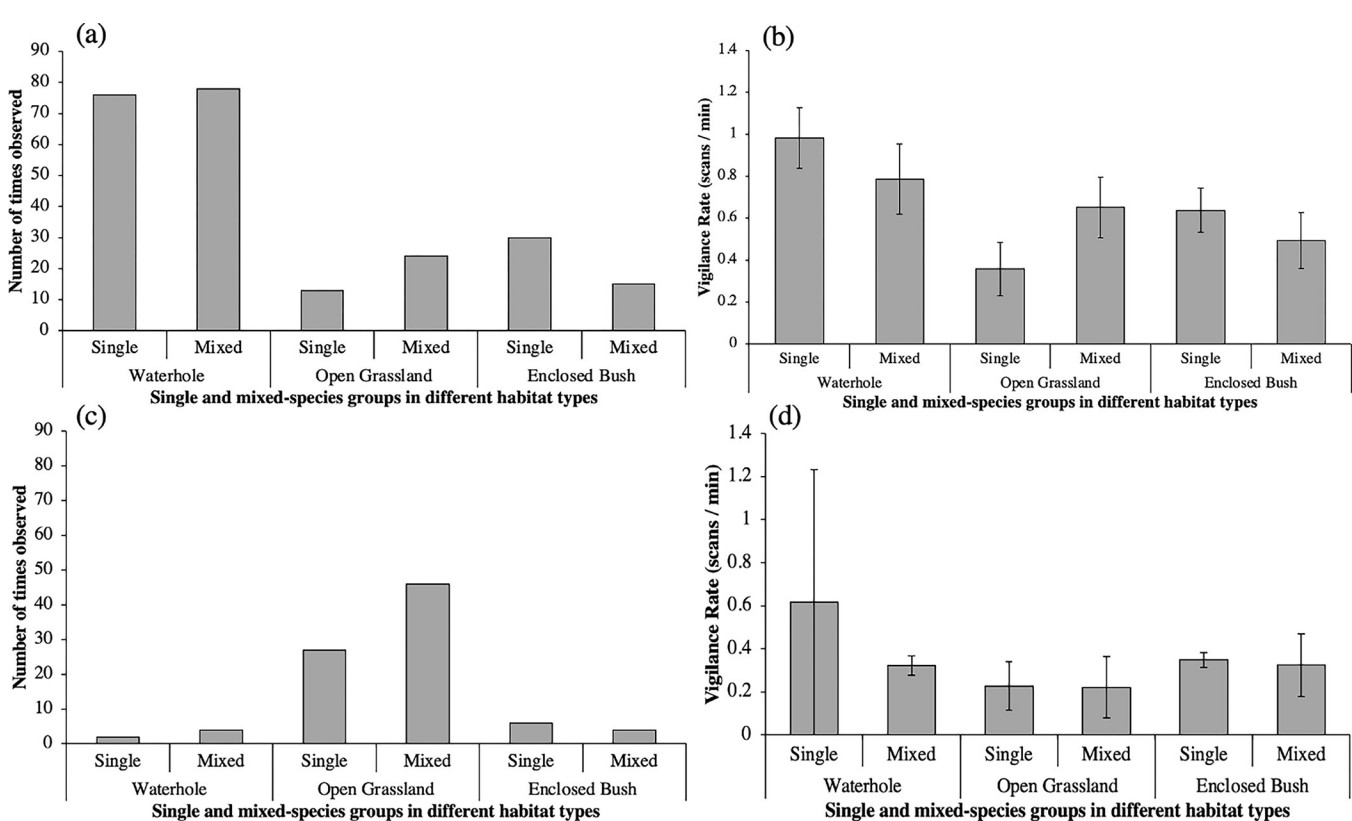

**Fig 6.** (a) Number of times animals were observed in single and mixed species groups within each habitat type, and (b) rates of vigilance displayed across these different environments.

**Table 10. Coefficients and P values for the final analysis on how habitat type influenced the formation of mixed-species groups in the wild (N = 237).**

| Fixed Effect | Estimate | Standard Error | P value |
|---|---|---|---|
| **Intercept** (waterhole) | 0.026 | 0.161 | 0.872 |
| **Habitat Type** | | | |
| Open Grassland | 0.513 | 0.372 | 0.169 |
| Vegetative Cover | -0.719 | 0.355 | 0.043* |

addition, the average species density and diversity during observations across wild and zoo environments highlighted similarities in how animals are associating and forming groups. These findings suggest that animals gain significant benefits from associating with each other regardless of setting. Previous research has shown that the formation of mixed species groups to be less likely when predators are absent [23], and the extent of the group size effect on vigilance to be reduced [20]. This would explain why there was no impact of mixed species grouping on vigilance within the zoo. However, as mixed species associations are still occurring at high rates within captivity, there are likely to be other drivers for this. Another main driver suggested for mixed species grouping in the wild is for increased foraging efficiency [18]. Animals were often observed feeding when in mixed species groups (on average 30.7% of the time) and hence optimal foraging may have been a key reason for mixed species assemblages. However, animals were also observed performing other activities such as resting for a large proportion of the time they were together (11.4%), during which time they may feel safer in a mixed species group. Advancing our understanding of the environmental conditions that influence which species will associate maybe particularly important for understanding how such species mixes will translate (harmoniously) to a zoo's enclosure. For example, affiliative or neutral mixed species social interactions may occur when multiple foraging areas are provided and therefore the value of a resource is diluted.

The study highlighted significant differences between a species' choice to form a mixed grouping with another specific species even when multiple species shared the same physical environment (could therefore have associated together). Certain species were found significantly more often in a mixed species group; giraffe, for example, were observed in the same group as oryx ten times in the zoo but never in a group with waterbuck (although these species were often all visible in the same areas of the enclosure). As oryx and various other species were not observed in the wild, we cannot suggest whether the association between giraffe and oryx would generalise to other groups of those species in the wild. Only six out of the 29 species observed were seen in both wild and in captivity (namely giraffes, white rhinoceros, impala, waterbuck, Cape buffalo and wildebeest), therefore this is a limitation to consider for most species-specific associations as we do not have a wide species diversity for analysis. Similarly, at Tembe Elephant Park, impala, nyala, and elephants were all observed to use the waterhole at similar times of the day. However, whilst nyala and impala were observed in the same group 16 times, nyala and elephants were observed in the same group eight times, and impala with elephants only twice. This suggests that these three species were tolerant to each other due to often sharing the same resource at the same time of day, but that impala may show some level of avoidance of proximity to elephants and a preference to spending time in proximity to nyala. Due to the similar ecology and size of nyala and impala, they may benefit from associating with each other, whereas they may need to avoid larger species [52] such as elephants to avoid aggression which has been previously observed to affect impala [53]. Impala display significant flexibility in their feeding habits [54], which may explain why they are able

to form mixed groups with many species different species without this creating too much competition. Walther [55] suggested that species which ignore each other may be best suited to be within the same enclosure, as this might avoid aggressive interactions compared to species which actively interact. Species that are used to sharing the same space but do not spend much time in proximity due to ignorance (also as opposed to avoidance) may be the most suitable for being housed together in captivity, where the lack of predators and foraging pressures make forming mixed species groups less important. It would therefore be beneficial for a further study to collect data on the affiliative and aggressive interactions between species to provide support for this suggestion and to distinguish species which are avoiding versus ignoring each other. This study demonstrates how both shared habitat and shared group are important factors to obtain a clearer picture of the extent to which different species tolerate and respond to each other within a confined zoo enclosure.

## 4.2. Vigilance

In the wild, vigilant behaviours were significantly influenced by various aspects of the social environment. The rate of vigilance decreased as the total number of animals present in the environment increased, and the proportion of time spent vigilant decreased with increasing species density. However, both vigilance rate and the proportion of time spent vigilant also increased with increasing species diversity. This supported our hypothesis and previous findings that animals display lower rates of vigilance when they are observed in a larger group, due to an increased safety from predation [20]. The group size in this study included all species present, suggesting that both conspecifics and heterospecifics contributed to a heightened feeling of safety from predation. However, these data suggest that when species diversity increased, the presence of heterospecifics caused vigilance to also increase, potentially due to increased need for social vigilance to keep track of many different species and individuals. Fig 2 shows that vigilance rates decreased when species diversity reached between three to five species, but then increased from six to eight species. Species diversities of six to eight were observed much less frequently than lower counts (a species diversity of seven and eight were only observed three times each), as indicated by the size of the error bars. The reason that animals are rarely observed in aggregations is a potential area for further study to define the social and environmental factors that leads to increased vigilance with increasing species diversity in a group. Zoos must consider the impact of different and optimal levels of species diversity for different species and how this may impact their behaviour when they are managed in a mixed species enclosure. Knowing "what is optimal" in terms of species diversity within a group for each specific named species is a crucial next step for this research thread.

In captivity, there was no significant impacts of group composition on vigilant behaviours. Vigilance rate of captive animals were much lower than in the wild (0.25 scans per minute compared to 0.77 scans per minute) which is mostly likely to due to the lack of predators in captivity and potentially explains a lack of strong effect from group composition. The results suggest that the positive impact of mixed species groups on vigilance are more relevant to an environment with predators (i.e., in the wild and not in captivity). However, as animals are still found to spend the same amount of time in a mixed species group in captivity as in the wild, therefore they are likely to still be gaining some benefits from this association.

Habitat characteristics additionally affected vigilance, with rates highest at waterholes compared to open grassland and bushes in both wild and zoo environments. This may be expected as a large proportion of attacks by predators occur at waterholes [56]. Smaller species also had higher rates of vigilance compared to medium or large-sized species, and birds had higher rates of vigilance than other mammals. Both findings are also expected because smaller

mammalian species and birds have been recorded as performing higher rates of vigilance in other previous research, possibly because they have more potential predators than larger species [40].

## 4.3. Other behaviours

The main similarities observed in the behaviour of wild and captive species included the amount of time spent feeding, vigilant, ruminating, and engaging in aggressive interactions. These are positive observations because feeding is an important behaviour that species in the wild spend a large proportion of time performing and this is a central aspect of captive animal management [8]. The natural rates of rumination observed is also a positive finding as this too is an important behaviour that ungulates perform when relaxed [57]. The fact that the proportion of time spent on vigilance and aggression was similar is also positive because it suggests that zoo-housed species may not be experiencing more distress between each other than they would in the wild. Low rates of aggression were observed in both wild and zoo environments during the study. The main differences observed were in the amount of time spent resting, locomoting, drinking, self-maintenance, and engaging in affiliative interactions. These differences would be expected because animals in captivity often have reduced activity levels [8] and have more spare time to engage in social interactions [58]. However, the difference in these behaviours was quite large (e.g., animals in zoos spent 28% of their time resting compared to 5.4% in the wild). Differences in resting, locomotion, and drinking may be explained by the fact that most observations in the wild were conducted at waterholes (where animals would be coming to drink and then moving away to safer areas), whereas in the zoo most observations were performed on open grassland. Wild animals are often more visible when performing certain behaviours over others, and less likely to be observable carrying out behaviour such as resting [32], which may bias data collected from the wild. Nonetheless, these data suggest that mixed species enclosures do help to increase locomotion behaviours in zoo animals, and additional management methods may be required to encourage a wider, beneficial array of active behaviours.

## 4.4. Social factors

Social factors (i.e., the sex composition of a group, and the presence of juveniles) did not have a significant influence on the likelihood of forming a mixed species group. When the observations were made, juveniles were already relatively mature, therefore we cannot rule out that these factors may have a stronger impact during mating and calving seasons. Certain patterns in our data are worthy of research extension to clarify what may be occurring. For example, Fig 3 shows a non-significant trend for all-female groups and mixed-sex groups to display lower vigilance rates in mixed species groups, compared to single species groups. Likewise, Fig 4 shows that all-female and mixed-sex groups were more likely to be observed in mixed species groups, whereas all-male groups were found equally in both. These data illustrated by Figs 3 and 4 are worthy of further investigation across more species and populations, and for a longer period of time, to ascertain any relevance to captive care and animal management moving forwards.

There was no difference in the rate of intraspecific aggression between single and mixed species groups. This goes against our predictions that were originally based on Leonardi, Buchanan-Smith [21] who found that intraspecific aggression decreased in the presence of other species when observing primates. Our dataset therefore suggests that this impact of mixed species grouping on intraspecific aggression may not be observable in ungulates. However, rates of aggression were very low during the study and thus running continuous sampling on aggressive behaviour may have been a more appropriate way of gathering these data.

## 4.5. Habitat type

Animals in the wild were found to spend significantly less time in a mixed species groups when observed within vegetative cover, compared to waterhole and grassland habitats (Fig 6). In the zoo there was no significant effect of habitat type, although this finding is may be limited because of the number of datapoints included in the logistic regression (89 datapoints per predictor instead of 100) and animals in the zoo were not observed frequently enough at the waterhole or under vegetative cover. We encourage further study across a longer time period to further understand these environmental impacts on social choice. The finding for wild data supports the suggestion that animals are observed to form more mixed species groups when there are higher risks of predation [23], which would therefore lead us to expect more mixed species groups near waterholes where predation levels are high [59], and vigilance was also highest during our study. Fig 6 shows that animals were also quite often observed in mixed species groups in open grassland where they are exposed and may benefit from forming mixed species groups. In contrast, in areas with outgrown vegetation, animals may be better concealed from predators when in smaller numbers and thus may not gain as much benefit from being in a mixed species group. This may be because a larger group may be more likely to be spotted by a predator [60], and that within vegetation it is more difficult for members to alert each other of danger [61] and to see when other individuals have become alert to danger [62]. These findings contrast with Beaudrot, Palmer [63] whose research found higher probabilities of mixed species groups in woodland habitat, which they suggested is due to increased risk of predation within woodland. Fig 6 shows no clear impact of mixed or single species groups on vigilance rates in different habitat types. This suggests that the impact of being in a mixed species group on vigilance does not vary significantly according to which habitat the animals are in be they wild or captive.

## 4.6. Species-specific findings

Individual species difference in vigilance maybe apparent between single and multiple species groups. Giraffe, waterbuck, zebra, and white rhinoceros showed reduced vigilance in a mixed species group, whereas data for impala, blue wildebeest, and nyala suggested no clear difference (Fig 3). Impala and nyala were frequently observed to associate in a mixed species group in the wild (Table 4), which could suggest some form of compatibility between these species. The presence of nyala also reduced the vigilance rates of impala (Fig 3). However, the presence of impala did not influence nyala vigilance rates, and nyala vigilance rates were not influenced by the presence of any other species. Impala were also frequently observed to associate with zebra, but with no clear impacts on impala vigilance rates. These apparent species-specific differences should be the subject of further observation to add more context and evaluation of why such patterns may appear.

Zebra were most frequently observed with impala and wildebeest and they displayed reduced vigilance in the presence of all species shown on the graph (Fig 3E). Zebra were also the only species observed to associate with warthogs and were seen in the same mixed group five times. Elephants were most often observed as a single species group but were found in a mixed species group with nyala eight times, and with woolly-necked storks five times. Defassa waterbuck were most frequently observed with nyala and showed reduced vigilance in the presence of nyala within the wild and Nile lechwe within the zoo. In the zoo, white rhinoceros were often observed in a group with Nile lechwe, and they also had lower rates of vigilance in all types mixed species groups (Fig 3F). In the zoo, giraffes were observed ten times with fringe-eared oryx and displayed reduced vigilance in their presence compared to when in a

single-species group, suggesting a potential benefit of the association between this species of antelope and the giraffe in this context.

## 4.7. Challenges and research extensions

This study shows a preliminary investigation which could be furthered in several ways. Based on the experiences of the observer, the Safari Park camera appeared to be targeted at larger, charismatic species (e.g., white rhinoceros, giraffe, and Cape buffalo). This meant that it was difficult to evenly collect data across all species in the enclosure. A longer period of study, coupled with direct observation, would be useful to further understand motivations for mixed- or single-species groupings with such an expansive captive environment. Additionally, access to more cameras within mixed species zoo enclosures would increase the number of individuals observed in captivity to increase generalisability of results [64]. It may additionally be useful to use a more refined protocol for evaluating which species are within the same group, such as using a ruler on the screen to measure distances between species.

It would be beneficial to have data on the presence of predators, e.g., by utilising data from researchers who are tracking predator's movements within the areas studied. This study could also be developed further by studying the impact of other important environmental stressors in addition to the presence of predators. In captivity, this could include comparing the effect of mixed species grouping on vigilance during times of high and low visitor numbers [65], or by the presence of zoo staff (e.g. veterinary professionals or caregivers). A comparison of species associations and corresponding behavioural changes across both day and night would allow for a wider understanding of any interrelationships [66]. Wider temporal and seasonal study across different times of the year (e.g. wet and dry seasons in the wild, or periods of more intense management in the zoo) would also be beneficial. Access to cameras within the wild ranges for more of the species observed within the zoo would be useful to allow for more direct comparisons of the same species across these two different (wild and captive) environments.

A future study should combine its findings with our data on interspecies associations to investigate whether tolerance as evidenced by close proximity in this study would predict more affiliative and/or neutral interspecific interactions and fewer aggressive interspecific interactions. Or alternatively whether increased proximity of species instead increases the chance of interspecific aggressive interactions. Valeix, Chamaillé-Jammes [53] observed that species such as impala that actively avoided elephants at waterholes were also those species more likely involved in aggressive interactions with elephants, whereas Cape buffalo, which did not avoid elephants, rarely engaged in conflict behaviour. Further consideration of a species' functional traits–ecological and morphological characteristics that influence an organism's fitness and performance–and application of methods and approaches from functional ecology–choosing relevant functional traits to describe the complementary functions of organisms that determine their ecosystem roles [67] could be a future avenue of extension of such research. There is application of functional trait knowledge to animal husbandry development and appropriate housing for specific species [68]; for example, the need of species to be vigilant whilst foraging when alone compared to when mixing with a group of another species of animal.

## 5. Conclusion

This study has provided useful, preliminary findings on the impact of mixed species enclosures and whether these replicate natural social environments and support natural species associations. Overall, the study found similar association patterns and group compositions in the wild and zoo. In the wild, these associations impacted vigilance rates in both positive and negative

ways (with numbers of animals decreasing vigilance but too much diversity increasing it). Whilst zoo-housed species did not display these effects on vigilance, they were still associating with other species at a similar rate, suggesting that they may still gain benefits from these associations in captivity. The study highlights the importance of differentiating species that can share the same habitat from species that chose to spend time within a mixed species group. These data can hopefully be used to stimulate further study to provide more evidence for housing different species in a way that contributes to optimal welfare for all species involved by assessing species' ecology and behaviour patterns when determining multi-species combinations.

## Supporting information

**S1 File.**
(DOCX)

**S2 File.**
(XLSX)

## Author Contributions

**Conceptualization:** Paul E. Rose.

**Data curation:** Claire Gauquelin Des Pallieres.

**Formal analysis:** Claire Gauquelin Des Pallieres.

**Investigation:** Claire Gauquelin Des Pallieres, Paul E. Rose.

**Methodology:** Claire Gauquelin Des Pallieres, Paul E. Rose.

**Project administration:** Claire Gauquelin Des Pallieres.

**Supervision:** Paul E. Rose.

**Writing – original draft:** Claire Gauquelin Des Pallieres, Paul E. Rose.

**Writing – review & editing:** Claire Gauquelin Des Pallieres, Paul E. Rose.

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
