## [Decision Letter · Decision Letter 0]

13 Dec 2022

PONE-D-22-30448Two’s company, three species is a crowd? A webcam-based study of the behavioural effects of mixed-species groupings in the wild and the zoo.PLOS ONE

Dear Dr. Rose,

Thank you for submitting your manuscript to PLOS ONE. After careful consideration, we feel that it has merit but does not fully meet PLOS ONE’s publication criteria as it currently stands. Therefore, we invite you to submit a revised version of the manuscript that addresses the points raised during the review process.

We look forward to receiving your revised manuscript.

Kind regards,

Carlos Rouco, PhD

Academic Editor

PLOS ONE

Journal Requirements:

4. Please upload a new copy of Figure 4 as the detail is not clear. Please follow the link for more information:

https://blogs.plos.org/plos/2019/06/looking-good-tips-for-creating-your-plos-figures-graphics/

https://blogs.plos.org/plos/2019/06/looking-good-tips-for-creating-your-plos-figures-graphics/

**Additional Editor Comments:**

Dear authors,

Since it has been hard to find potential reviewers for the ms I did myself reviewed it. I think that is an original idea and method to survey and compare behaviours between wildlife and captive animals. However, I agree with the referee that this paper, in its current setup, may not give much information that is useful within zoos. I also agree with the referee that the discussion is over-reached – e.g. how this information can be used directly in terms of managing zoos? In addition, it is hard to compare the behaviour of wild animal, with captive ones, the latest have less predation risk, food availability, no harsh weather conditions, within others… I suggest authors to erect alternative hypotheses to their main conclusion, i.e. “The results suggest that these species benefit from increased perceived safety in larger groups, regardless of the species making up that group”, accounting for the main differences between wild and captive species behaviour. As rose by the referee, the way the data is presented in terms of combined zoo and wild data may be masking lots of differences, and it may be an error to join them together (due to the differences between the environments but principally the fact that zoos are keeping animals in pre-defined groups, these associations aren’t a choice). I would also encourage the authors to re-do the analyses with the two groups (zoo vs wild) as separate entities, and then compare them descriptively in the discussion. I would also encourage authors to look at fuller activity budgets rather than just vigilance. Authors should also state much clearly on the rationale for solely including vigilance since its application in zoos is relatively limited, especially as high levels of vigilance in the zoo environment it is not desirable. And finally, they should state much clearly, how their research may be useful or may be applied in zoos.

Carlos Rouco

Academic Editor

Reviewers' comments:

Reviewer's Responses to Questions

**Comments to the Author**

1. Is the manuscript technically sound, and do the data support the conclusions?

Reviewer #1: Partly

2. Has the statistical analysis been performed appropriately and rigorously? 

Reviewer #1: Yes

3. Have the authors made all data underlying the findings in their manuscript fully available?

Reviewer #1: Yes

4. Is the manuscript presented in an intelligible fashion and written in standard English?

Reviewer #1: Yes

5. Review Comments to the Author

Reviewer #1: Dear authors,

Thank you for submitting this paper, I thought it was an interesting piece of work and is quite a nice idea in terms of understanding a little more about mixed species associations in the wild and considering how that can be transferred over to zoos.

I did find some aspects of the work a little confusing and I think it needs some work to make the findings clearer and more applicable. My main comment is that I think you need to separate out the zoo and wild data for analysis. These two environments are so completely different that I think by combining them you are muddying the waters and I don’t think the conclusions hold valid because of it. The zoo is of course an enforced mixed species exhibit and an exhibit with no natural predators so is a completely different environment to the wild, and I think that is why they need to be treated separately. I am also unclear why you chose to focus only on vigilance, when you collected a whole activity budget. Vigilance is something we really drive hard not to have excessive amounts of in a zoo, and I think comparing whole activity budgets would be much more beneficial in terms of the aims of your study.

General comments on methods:

More detail needed on the zoo enclosure – approx. how big is the space and how much of that space was covered by the camera

Habitat type is confusing, is this wild animals only? How are you defining different types of habitat within the zoo?

L263/264: why did you use chi squared here? This data doesn’t appear to be categorical so a t-test or mann whitney would have been more appropriate

Your factors in relation to ‘predicted likelihood of forming mixed social grps’ don’t make sense for zoos – the grps are formed by the zoo, there is nothing to predict it. you can look at whether they predict proximity but that is slightly different to what I think you are suggesting

I don’t think within 6 body lengths is appropriate for a zoo setting – that’s a pretty large distance from another animal, I think you need a strong rationale for why you chose that and not for example within one or two body lengths as would normally be done

You need details on number of grps where you did know the sexes of animals as the number of these will show whether your results in relation to this are affected by small samples

Information needed on the zoo in terms of numbers of species/diversity of sp in the mixed sp enclosure

General comments on discussion:

Discussion feels a bit muddled and like it has lost its way a little bit – I think you need to split out zoo and wild data and then consider more what we can do with the wild data to help support zoo management decisions. I also think the discussion will be stronger if you look at activity budgets more fully.

Line by line comments:

L40: give examples of habitat type

L47/48: I would argue that increased vigilance is not something

L85: ‘have much potential’ is a bit clunky – suggest rewording

L88: maybe, you can’t guarantee these risks can be avoided

L92: enclosures

L95: whether these associations change and if they influence… - you don’t know they definitely do – also, is this referring to zoo or wild? Needs clarity throughout on the two environments

L108/109: typos

L113: provide some examples of sp which are commonly kept in mixed sp grpings

L115: in the wild? Zoo?

L124: not sure how you are answering Q 3?

L125: Q #4- you are forcing the sp associations within a zoo so I think this needs to refer to spatial associations

L130: what do you mean by ‘particularly close proximity’?

L134: six body lengths is huge in a zoo – needs strong rationale

L134: if animals are forced to associate because of resources how do you know it is a choice to aggregate how they are?

L149: you were only looking at vigilance so hypothesis #5 feels the opposite of hypothesis #4

L252: it is not clear if this is wild animals only, although I am not clear how it fits with the zoo environment, needs to specify

L308: see comments above – I don’t think it makes sense to combine the data

L310: not sure what is meant by ‘this was not significant for individual sp either’?

L313: is this wild or zoo? need to clarify

Tbl 5: make subtitles in the table bold so it’s clearer

L380: not sure what the n=150 refers to?

L381: you don’t know it is significantly influencing it – it stands to reason that larger grps will consist of more species but that’s not to say that correlation = causation here. Be mindful of this in interpretation through the discussion too.

L397: how does this relate to zoos?

L422 – 424: not sure how this has been reached from the data presented?

L450: reduced vigilance as compared to what? Giraffe on their own? Or comparison to the wild?

L458 – 461: it’s quite likely predators were just off camera. Predators in the wild are driving associations (amongst other things) so they’re a big driver of social grps

L464 – 468: you haven’t included this information in your results – but I do think you need to

L469 – 471: Yes! Which is what makes this study quite confusing

L481: this is a strong rationale for analysing this data separately and only comparing where you have similar mixes between wild and zoo environments

L528/529: they surely wouldn’t do it if it did create stress. There could be multiple other environmental reasons causing this.

L530: how did sp diversity compare in zoo/wild?

L537: why do they perform more vigilance?

L550: needs a citation – not sure how/why this is considered more energetically costly than vigilance as a state behaviour?

L576: how many grps are sexually dimorphic? How representative is this?

L583: how does this compare with data from grps of ungulates

L635: you could look at this with your data – and I would encourage you to look at this with your data

L649: you only looked at vigilance

L654 – 656: I don’t think you can say this as you’ve not really got any idea why vigilance reduced

6. PLOS authors have the option to publish the peer review history of their article (what does this mean?). If published, this will include your full peer review and any attached files.

Reviewer #1: No

---

## [Author Response · Author response to Decision Letter 0]

1 Mar 2023

Replies to editor:

Dear authors,

Since it has been hard to find potential reviewers for the ms I did myself reviewed it. I think that is an original idea and method to survey and compare behaviours between wildlife and captive animals. However, I agree with the referee that this paper, in its current setup, may not give much information that is useful within zoos. I also agree with the referee that the discussion is over-reached – e.g. how this information can be used directly in terms of managing zoos? In addition, it is hard to compare the behaviour of wild animal, with captive ones, the latest have less predation risk, food availability, no harsh weather conditions, within others… I suggest authors to erect alternative hypotheses to their main conclusion, i.e. “The results suggest that these species benefit from increased perceived safety in larger groups, regardless of the species making up that group”, accounting for the main differences between wild and captive species behaviour. As rose by the referee, the way the data is presented in terms of combined zoo and wild data may be masking lots of differences, and it may be an error to join them together (due to the differences between the environments but principally the fact that zoos are keeping animals in pre-defined groups, these associations aren’t a choice). I would also encourage the authors to re-do the analyses with the two groups (zoo vs wild) as separate entities, and then compare them descriptively in the discussion. I would also encourage authors to look at fuller activity budgets rather than just vigilance. Authors should also state much clearly on the rationale for solely including vigilance since its application in zoos is relatively limited, especially as high levels of vigilance in the zoo environment it is not desirable. And finally, they should state much clearly, how their research may be useful or may be applied in zoos.

Thank you for the comments. We are pleased that some aspects of this paper are deemed useful and we have been grateful for the opportunity for revisions. We have actioned all comments and edits by the reviewer and editor.

Replies to reviewer 1

Reviewer #1: Dear authors,

Thank you for submitting this paper, I thought it was an interesting piece of work and is quite a nice idea in terms of understanding a little more about mixed species associations in the wild and considering how that can be transferred over to zoos.

I did find some aspects of the work a little confusing and I think it needs some work to make the findings clearer and more applicable. My main comment is that I think you need to separate out the zoo and wild data for analysis. These two environments are so completely different that I think by combining them you are muddying the waters and I don’t think the conclusions hold valid because of it. The zoo is of course an enforced mixed species exhibit and an exhibit with no natural predators so is a completely different environment to the wild, and I think that is why they need to be treated separately. I am also unclear why you chose to focus only on vigilance, when you collected a whole activity budget. Vigilance is something we really drive hard not to have excessive amounts of in a zoo, and I think comparing whole activity budgets would be much more beneficial in terms of the aims of your study.

I have now separated out the analysis for zoo and wild data, and I have also included further information on a wider variety of behaviour types.

General comments on methods:

More detail needed on the zoo enclosure – approx. how big is the space and how much of that space was covered by the camera.

I have added more information on the zoo enclosure.

Habitat type is confusing, is this wild animals only? How are you defining different types of habitat within the zoo?

It is used for both the zoo and the wild individuals and I have clarified this in the text and explained why it was also used in the zoo.

L263/264: why did you use chi squared here? This data doesn’t appear to be categorical so a t-test or mann whitney would have been more appropriate

I have re-phrased this paragraph to make more sense as a chi-squared test.

Your factors in relation to ‘predicted likelihood of forming mixed social grps’ don’t make sense for zoos – the grps are formed by the zoo, there is nothing to predict it. you can look at whether they predict proximity but that is slightly different to what I think you are suggesting

In the study I labelled a 'group' as being within six body lengths of each other, to differentiate animals that are just in the same area from animals that are potentially associating. I explained this in the methods but have also now clarified this further now so that it hopefully makes more sense. It is essentially about predicting proximity, as you suggest, as in the study my definition of a group is purely based on the extent of proximity between individuals. For example, whilst white white rhinoceroses and Thompson’s gazelles are both in the same enclosure, they were never observed to be anywhere near six body lengths from each other, and so were never labelled as being in the same group.

I don’t think within 6 body lengths is appropriate for a zoo setting – that’s a pretty large distance from another animal, I think you need a strong rationale for why you chose that and not for example within one or two body lengths as would normally be done.

I hope that my further detail on the size of the enclosure might help to explain why we felt that six body lengths was appropriate.

You need details on number of grps where you did know the sexes of animals as the number of these will show whether your results in relation to this are affected by small samples.

I have added information in the methods section and results.

Information needed on the zoo in terms of numbers of species/diversity of sp in the mixed sp enclosure.

I have added this information in the methods.

General comments on discussion:

Discussion feels a bit muddled and like it has lost its way a little bit – I think you need to split out zoo and wild data and then consider more what we can do with the wild data to help support zoo management decisions. I also think the discussion will be stronger if you look at activity budgets more fully.

These changes have been made and the discussion is now more fluid.

Line by line comments:

L40: give examples of habitat type.

I have provided examples.

L47/48: I would argue that increased vigilance is not something.

I have removed this section.

L85: ‘have much potential’ is a bit clunky – suggest rewording.

I have re-worded this sentence.

L88: maybe, you can’t guarantee these risks can be avoided.

I have re-worded to highlight this.

L92: enclosures.

Resolved.

L95: whether these associations change and if they influence… - you don’t know they definitely do – also, is this referring to zoo or wild? Needs clarity throughout on the two environments.

I have clarified this.

L108/109: typos.

Resolved.

L113: provide some examples of sp which are commonly kept in mixed sp grpings

I have added a relevant citation.

L115: in the wild? Zoo?

Re-phrased.

L124: not sure how you are answering Q 3?

I have re-worded this question to make more sense with the analysis done.

L125: Q #4- you are forcing the sp associations within a zoo so I think this needs to refer to spatial associations.

I felt that the location of the cameras near water sources in the wild locations created a somewhat similar situation to within the zoo enclosure, where animals are to some extent forced to use the same space (in the zoo because they are in an enclosure, and in the wild because they are all using the same resource), however I aimed to use the six body lengths to identify species that were closer than required even whilst using the same resource, both at the zoo and in the wild spaces.

L130: what do you mean by ‘particularly close proximity’?

I have clarified what is meant here by referring directly to the six body lengths measurement used.

L134: six body lengths is huge in a zoo – needs strong rationale

I have added details of the size of this San Diego Zoo enclosure. It is a large space, roughly 12 hectares in size, within which it is very easy for animals to stay at a significant distance to each other. Many species in this enclosure were never observed within six body lengths of each other. 

L134: if animals are forced to associate because of resources how do you know it is a choice to aggregate how they are?

Please see response to comment L125.

L149: you were only looking at vigilance so hypothesis #5 feels the opposite of hypothesis #4

I have now included analyses of additional behaviours rather than just looking at vigilance.

H4 says that overall vigilance will be lower, but doesn’t suggest an impact of grouping, whereas H5 is focussing on the impact of grouping, and saying that vigilance may be even lower when in a mixed species group in captivity.

L252: it is not clear if this is wild animals only, although I am not clear how it fits with the zoo environment, needs to specify

Please see response to comment L134. The zoo enclosure is a very large space with a waterhole, areas of vegetative cover, and areas of open grass.

L308: see comments above – I don’t think it makes sense to combine the data.

I have changed the analyses to look at zoo and wild data separately.

L310: not sure what is meant by ‘this was not significant for individual sp either’?

I have re-worded this to be clearer. 

L313: is this wild or zoo? need to clarify

Re-phrased.

Tbl 5: make subtitles in the table bold so it’s clearer

Done.

L380: not sure what the n=150 refers to?

I have clarified this.

L381: you don’t know it is significantly influencing it – it stands to reason that larger grps will consist of more species but that’s not to say that correlation = causation here. Be mindful of this in interpretation through the discussion too.

I have removed focal group size from the analysis as I realise it wasn't suitable.

L397: how does this relate to zoos?

Please see response to comment L134 & L252. 

L422 – 424: not sure how this has been reached from the data presented?

Line deleted.

L450: reduced vigilance as compared to what? Giraffe on their own? Or comparison to the wild?

I have clarified this statement.

L458 – 461: it’s quite likely predators were just off camera. Predators in the wild are driving associations (amongst other things) so they’re a big driver of social grps

I have now removed the comment about the presence of predators not significantly influencing the associations and have made a comment about this limitation in the limitations section.

L464 – 468: you haven’t included this information in your results – but I do think you need to

I have now included additional information about general activity other than vigilance.

L469 – 471: Yes! Which is what makes this study quite confusing

I have tried to explain the way I was looking at associations and mixed species groupings more clearly, please see above my comments on six body lengths and this distinction of a group.

L481: this is a strong rationale for analysing this data separately and only comparing where you have similar mixes between wild and zoo environments

I have done this now.

L528/529: they surely wouldn’t do it if it did create stress. There could be multiple other environmental reasons causing this.

Noted.

L530: how did sp diversity compare in zoo/wild?

I have added information on this in the results and discussion.

L537: why do they perform more vigilance?

I have added an explanation for this.

L550: needs a citation – not sure how/why this is considered more energetically costly than vigilance as a state behaviour?

Comment removed after change to analyses.

L576: how many grps are sexually dimorphic? How representative is this?

I have added information on this in the methods and results.

L583: how does this compare with data from grps of ungulates

Re-phrased.

L635: you could look at this with your data – and I would encourage you to look at this with your data.

I didn't collect data on inter-species interactions, only intra-species interactions, as there was already a lot of data to collect and I found it easier to only have to focus on the behaviour of the focal species.

L649: you only looked at vigilance

We now include other behaviours.

L654 – 656: I don’t think you can say this as you’ve not really got any idea why vigilance reduced

Removed.

---

## [Decision Letter · Decision Letter 1]

14 Mar 2023

PONE-D-22-30448R1Two’s company, three species is a crowd? A webcam-based study of the behavioural effects of mixed-species groupings in the wild and in the zoo.PLOS ONE

Dear Dr. Rose,

Thank you for submitting your manuscript to PLOS ONE. After careful consideration, we feel that it has merit but does not fully meet PLOS ONE’s publication criteria as it currently stands. Therefore, we invite you to submit a revised version of the manuscript that addresses the points raised during the review process.

We look forward to receiving your revised manuscript.

Kind regards,

Carlos Rouco, PhD

Academic Editor

PLOS ONE

Journal Requirements:

Additional Editor Comments :

Dear Authors,

I have received now the comment from the reviewer and I am agree that the Ms has improved. I will recommend to you to follow all suggestions rose by reviewer, and pay special attention to the assertion left in the discussion (L651 – 653), which I don’t think you can state from your work, therefore I would encourage you to consider moving it away from discussing ‘trends’ in the data set, especially when you rightly recommended limitations in the analytical methods/the data.

Carlos Rouco

Academic Editor

Reviewers' comments:

Reviewer's Responses to Questions

**Comments to the Author**

1. If the authors have adequately addressed your comments raised in a previous round of review and you feel that this manuscript is now acceptable for publication, you may indicate that here to bypass the “Comments to the Author” section, enter your conflict of interest statement in the “Confidential to Editor” section, and submit your "Accept" recommendation.

Reviewer #1: (No Response)

2. Is the manuscript technically sound, and do the data support the conclusions?

Reviewer #1: Yes

3. Has the statistical analysis been performed appropriately and rigorously? 

Reviewer #1: Yes

4. Have the authors made all data underlying the findings in their manuscript fully available?

Reviewer #1: Yes

5. Is the manuscript presented in an intelligible fashion and written in standard English?

Reviewer #1: Yes

6. Review Comments to the Author

Reviewer #1: Dear authors,

Thank you for taking the time to respond to my review and make subsequent modifications to the manuscript. I think the separation into zoo/wild has made it much easier to follow, and actually has led to some useful comparisons.

I have just a few minor comments:

L44: might be worth adding in a sentence about zoos here – i.e. maybe no impact because of the reduced need for animals to show heightened vigilance

L291/292: why did you run the test for just some of the sp individually, I think this needs to be all or nothing, unless there is a strong rationale for just selecting some

Table 3: you have four species here but at L291/292 you just said giraffe and waterbuck

L532/533: I am not sure what this first sentence means

L618: there is a word missing before ‘experiencing’, maybe ‘be’?

L641 – 643: I don’t particularly think it is useful to discuss trends, as you don’t want people to take too much from it

L264 – 265: is this another trend or is it statistically significant?

L651 – 653: this feels like quite a strong assertion – especially given you have said a few lines before that males spent equal time in mixed and non-mixed groups. I’m concerned this comment might lead to even less interest in bachelor grps, whereas in actual fact we should be doing more than ever to make sure welfare of surplus males is good

L669: remove ‘so’

L670 – 671: this sort of begs the question of why this analysis was done

L689: is this section wild animals, or zoo animals, or both? Needs clarifying

L691: what do you mean ‘particularly strong trends’?

7. PLOS authors have the option to publish the peer review history of their article (what does this mean?). If published, this will include your full peer review and any attached files.

Reviewer #1: No

---

## [Author Response · Author response to Decision Letter 1]

19 Mar 2023

Editor comment

I have received now the comment from the reviewer and I am agree that the Ms has improved. I will recommend to you to follow all suggestions rose by reviewer, and pay special attention to the assertion left in the discussion (L651 – 653), which I don’t think you can state from your work, therefore I would encourage you to consider moving it away from discussing ‘trends’ in the data set, especially when you rightly recommended limitations in the analytical methods/the data.

Thank you for the comment. We are pleased that the manuscript is now improved. We have deleted the text in lines 651-653 and we have reviewed the discussion to ensure that trends and patterns are made clear and not passed off as significance. We have clarified sections of the discussion to remove ambiguity and ensured that figures are specifically linked to descriptions of the animals’ behaviour. 

Reviewer comment

Thank you for taking the time to respond to my review and make subsequent modifications to the manuscript. I think the separation into zoo/wild has made it much easier to follow, and actually has led to some useful comparisons.

Thank you for the useful and developmental feedback. We appreciate your time and effort in helping shape this manuscript into a more relevant piece of work.

I have just a few minor comments:

L44: might be worth adding in a sentence about zoos here – i.e. maybe no impact because of the reduced need for animals to show heightened vigilance

Thank you for the useful edit. We have included this suggestion.

L291/292: why did you run the test for just some of the sp individually, I think this needs to be all or nothing, unless there is a strong rationale for just selecting some

Thank you for the comment. This was just the Chi squared for the number of times observed in mixed and single species groups for four example species that were seen in the wild and in the zoo and that may experience a markedly different social environment in the zoo (because of their wild ecology – group number and size, dispersal or distance between individuals) to check for any potential bias or outliers in the species under study. We have included further references to explain why these species were selected. 

Table 3: you have four species here but at L291/292 you just said giraffe and waterbuck

Edited for clarity in the methods.

L532/533: I am not sure what this first sentence means

We have edited this section for clarity.

L618: there is a word missing before ‘experiencing’, maybe ‘be’?

Thank you for the correction. We have also edited this section for clarity.

L641 – 643: I don’t particularly think it is useful to discuss trends, as you don’t want people to take too much from it

We have re-written this as a call for more research to find out why this trend might occur.

L264 – 265: is this another trend or is it statistically significant?

We are not sure what you mean here as these lines are in the methods section? Please can this be clarified. 

L651 – 653: this feels like quite a strong assertion – especially given you have said a few lines before that males spent equal time in mixed and non-mixed groups. I’m concerned this comment might lead to even less interest in bachelor grps, whereas in actual fact we should be doing more than ever to make sure welfare of surplus males is good

This, and the section in lines 641-643 have been completely re-written. If the reviewer is still unhappy, we can delete this section, but we feel it is worthwhile area of research extension to encourage others to look into.

L669: remove ‘so’

Edited 

L670 – 671: this sort of begs the question of why this analysis was done

We have clarified this section further.

L689: is this section wild animals, or zoo animals, or both? Needs clarifying

Edited for clarity. 

L691: what do you mean ‘particularly strong trends’?

Removed and edited for clarity

---

## [Editor Report · Decision Letter 2]

27 Mar 2023

Two’s company, three species is a crowd? A webcam-based study of the behavioural effects of mixed-species groupings in the wild and in the zoo.

PONE-D-22-30448R2

Dear Dr. Rose,

We’re pleased to inform you that your manuscript has been judged scientifically suitable for publication and will be formally accepted for publication once it meets all outstanding technical requirements.

Kind regards,

Carlos Rouco, PhD

Academic Editor

PLOS ONE

---

## [Editor Report · Acceptance letter]

13 Apr 2023

PONE-D-22-30448R2 

Two’s company, three species is a crowd? A webcam-based study of the behavioural effects of mixed-species groupings in the wild and in the zoo 

Dear Dr. Rose:

I'm pleased to inform you that your manuscript has been deemed suitable for publication in PLOS ONE. Congratulations! Your manuscript is now with our production department. 

Kind regards, 

on behalf of

Dr. Carlos Rouco 

Academic Editor

PLOS ONE